# Use of malaria rapid diagnostic test and anti-malarial drug prescription practices among primary healthcare workers in Ebonyi state, Nigeria: An analytical cross-sectional study

Ugwu I. Omale[1]*, Benedict N. Azuogu[1,2], Adaoha P. Agu[1,2], Edmund N. Ossai[1,2]

1 Department of Community Medicine, Alex Ekwueme Federal University Teaching Hospital Abakaliki (AEFUTHA), Abakaliki, Ebonyi State, Nigeria, 2 Department of Community Medicine, College of Health Sciences, Ebonyi State University (EBSU), Abakaliki, Ebonyi State, Nigeria

* omaleiu@gmail.com

**Data Availability Statement:** All relevant data are within the manuscript and its Supporting information files.

## Abstract

### Background

The recommendation of universal diagnostic testing before malaria treatment aimed to address the problem of over-treatment with artemisinin-based combination therapy and the heightened risk of selection pressure and drug resistance and the use of malaria rapid diagnostic test (MRDT) was a key strategy, particularly among primary healthcare (PHC) workers whose access to and use of other forms of diagnostic testing were virtually absent. However, the use of MRDT can only remedy over-treatment when health workers respond appropriately to negative MRDT results by not prescribing anti-malarial drugs. This study assessed the use of MRDT and anti-malarial drug prescription practices, and the predictors, among PHC workers in Ebonyi state, Nigeria.

### Methods

We conducted an analytical cross-sectional questionnaire survey, among consenting PHC workers involved in the diagnosis and treatment of malaria, from January 15, 2020 to February 5, 2020. Data was collected via structured self-administered questionnaire and analysed using descriptive statistics and bivariate and multivariate generalized estimating equations.

### Results

Of the 490 participants surveyed: 81.4% usually/routinely used MRDT for malaria diagnosis and 18.6% usually used only clinical symptoms; 78.0% used MRDT for malaria diagnosis for all/most of their patients suspected of having malaria in the preceding month while 22.0% used MRDT for none/few/some; 74.9% had good anti-malarial drug prescription practice; and 68.0% reported appropriate response to negative MRDT results (never/rarely prescribed anti-malarial drugs for the patients) while 32.0% reported inappropriate response (sometimes/often/always prescribed anti-malarial drugs). The identified predictor(s): of the use of MRDT was working in health facilities supported by the United States' President's

**Funding:** The author(s) received no specific funding for this work.

**Competing interests:** The authors have declared that no competing interests exist.

Malaria Initiative (PMI-supported health facilities); of good anti-malarial drug prescription practice were having good opinion about MRDT, having good knowledge about malaria diagnosis and MRDT, being a health attendant, working in PMI-supported health facilities, and increase in age; and of appropriate response to negative MRDT results was having good opinion about MRDT.

## Conclusions

The evidence indicate the need for, and highlight factors to be considered by, further policy actions and interventions for optimal use of MRDT and anti-malarial drug prescription practices among the PHC workers in Ebonyi state, Nigeria, and similar settings.

## Introduction

Malaria is still causing significant morbidity and mortality particularly among children under 5 years old and pregnant women in high burden countries, including Nigeria [1], despite it being a preventable and treatable disease [1–3]. Nigeria has the highest burden of malaria in the world and accounted for about 27% of the global malaria cases and 31% of the global malaria deaths in 2021 [1] and Ebonyi state is one of the states in Nigeria with high burden of malaria as the prevalence of malaria parasitaemia among eligible children aged 6–59 months in the state was 25.7% in the 2021, the highest in the south-east geopolitical zone and higher than the national prevalence of 22.3% [4]. The major malaria parasite in Nigeria, and in Ebonyi state, is plasmodium (P.) *falciparum*. In 2021 in Ebonyi state, P. *falciparum* accounted for 90.8% of the infections in children under 5 years old while P. *malariae* accounted for 1.4%, P. *ovale* for 2.6%, and P. *falciparum* and P. *malariae* for 5.3% (mixed infection) [4].

Malaria diagnosis could occur via four ways: presumptively by the use of clinical symptoms, by malaria rapid diagnostic test (MRDT), by light microscopy, and by polymerase chain reaction (PCR) [5, 6]. In 2010, the World Health Organization (WHO) recommended that all patients suspected of having malaria receive prompt parasitological testing (with microscopy or MRDT) to confirm diagnosis before treatment [6]. The WHO recommendation for universal testing is based on some fundamental change in malaria trend worldwide such as the declining malaria incidence in high burden countries, the emergence of parasite resistance to anti-malarial drugs, especially artemisinin-based combination therapy (ACT), and the increased availability of diagnostic testing such as MRDT [7, 8]. The use of MRDT is a key part of the strategy for universal parasitological diagnostic testing recommended by the WHO [6, 7, 9, 10] as it is much feasible to deploy and use by PHC workers even in remote rural settings. Like many other countries, Nigeria and her foreign partners have scaled-up the availability and use of MRDT particularly by primary healthcare (PHC) workers who are the first points of contact (within the national healthcare system) for majority of the population (especially those living in the rural areas) [11] and universal parasitological diagnosis of malaria before treatment has been recommended by the Nigerian National Guidelines for Diagnosis and Treatment of Malaria [12, 13]. In this regard, the United States Agency for International Development President's Malaria Initiative (USAID PMI) provides support or supplies MRDT kits (and ACT) to many PHC facilities in Ebonyi state for free or subsidized malaria diagnostic (and treatment) services.

The recommendations of ACT as the first-line anti-malaria drug in 2006 [14] was followed by progressive and widespread increase in the use of ACT across Nigeria [15–17] and sub-

Saharan African countries [18, 19] that is associated with over-treatment of malaria with ACT and heightened risk of selection pressure and drug resistance. Though the recommendation of universal diagnostic testing before malaria treatment aimed to address this problem [6, 12, 13], many health workers in Nigeria still over-treated malaria with ACT as a result of inaccurate presumptive diagnosis (non-use of parasitological diagnosis) [20–25] and many still prescribed anti-malarial drugs/ACT when MRDT result was negative [26–28], with negative implications on the emergence of drug resistance. The use of MRDT can only remedy over-treatment of malaria (and the attendant higher risk of drug resistance) when health workers (and patients) respond appropriately to negative MRDT results by not prescribing anti-malarial drugs [29]. As was the case with the previous generations of anti-malarial drugs, the development and spread of malaria parasite resistance to ACT (the mainstay of malaria treatment) will result in significant increase in malaria mortality (especially in children) [30].

There is relative paucity of literature on the use of MRDT and anti-malarial drug prescription practices and their predictors. Although some studies in Nigeria had assessed the use of MRDT among health workers [23–25, 31], only few assessed the malarial drug use pattern [27, 28] and there was marked variation in the reported use of MRDT and anti-malarial drug prescription practices. Also, the use of MRDT, especially at the PHC level, has made universal access to parasitological diagnosis of malaria possible [5]. Since PHC workers operate mainly in the PHC facilities [32] that mostly serve the rural areas [11], where other form of parasitological diagnosis of malaria such as microscopy was virtually non-existent, there was need to investigate their use of MRDT and anti-malarial drug prescription practices.

The aim of this study was to assess the use of MRDT for malaria diagnosis and anti-malarial drug prescription practices, including appropriate response to negative MRDT results, and their predictors, among PHC workers in Ebonyi state, Nigeria, in order to generate empirical evidence to inform policy and programmatic interventions in Ebonyi state and similar settings.

## Methods

### Study design, setting and participants

This study was an analytical cross-sectional survey implemented between January 15, 2020 and February 5, 2020 in Ebonyi state and was part of a concurrent independent mixed method study among PHC workers. Ebonyi state is located in the south-east geopolitical zone of Nigeria and the projected population for 2020 was 3,251,508 based on the 2006 national census figure and a growth rate of 2.8% [33]. The people of Ebonyi are primarily of the Igbo ethnic extraction with ten dialects spoken across the state [34]. English, especially its local variant, the pidgin, is a widely spoken language in the state. Most inhabitants practice Christianity. Ebonyi state is divided into three senatorial zones (Ebonyi north, Ebonyi central and Ebonyi south) and 13 local government areas (LGAs). By the end of 2020, there were 784 health facilities in the state, consisting of 566 public facilities (2 Tertiary, 13 Secondary and 551 PHC facilities) and 218 private facilities (25 Secondary and 193 primary health facilities [35]. The tertiary and secondary healthcare facilities are mostly found in the urban areas while the rural areas are mostly served by the PHC facilities [11].

The PHC workers operate mainly in the public PHC facilities and the main categories of these workers include community health extension workers (CHEWs), community health officers (CHOs), nurses and midwives and health attendants. The CHEWs and CHOs are mid-level health workers in Nigeria trained to provide basic essential and public health services in PHC facilities and in the communities. The CHEWS, CHOs, and nurses and midwives receive formal trainings. Health attendants are health assistants or community resource persons

(CORPs) that receive informal training at PHC facilities and are allowed to treat minor ailments like uncomplicated malaria. Other categories of health workers with formal trainings such as environmental health officers, environmental health technicians, and medical lab technicians also work at the PHC facilities.

The study participants were the PHC workers in Ebonyi state, Nigeria. Eligible participants were those involved in the diagnosis and treatment of malaria at PHC facilities providing maternal and child healthcare services, including immunization, who were not on annual leave during the period of data collection, who were not on annual leave in the preceding one month to the period of data collection (because data about MRDT use in the preceding month was also collected), and who gave written consent.

## Sample size and sampling technique

Sample size of 220 was estimated using 95% level of confidence, 5% level of precision, 82.7% as the expected proportion (based on the proportion of public health workers that routinely use MRDT for the diagnoses of malaria in a previous study [31], and the sample size formula for single proportion. Provision was made for a response rate of 90% by dividing the calculated sample size of 220 by 0.9 to get 245. Because of multistage (cluster) sampling, the 245 was multiplied by a design effect of 2 to give the final sample size of 490 [36, 37].

Study participants were selected via multistage sampling technique. The LGAs in Ebonyi state were divided into three strata based on the three senatorial zones in the state and two LGAs were selected from each stratum by balloting. In each of the selected LGAs, the list of PHC facilities providing maternal and child healthcare services, including immunization, was obtained from the Ebonyi State Malaria Elimination Programme (SMEP) and consisted of 28 of such facilities in Izzi, 34 in Abakaliki, 30 in Ikwo, 24 in Ezza north, 31 in Afikpo north, and 27 in Afikpo south, giving a total of 174 PHC facilities. Formative assessment showed there were at least four clinical staff in each facility. Therefore, 123 (490/4) PHC facilities were required to achieve the sample size of 490 PHC workers. In each LGA PHC facilities were selected by balloting, using probability proportional to size, from the list of PHC facilities: for example, for Izzi, (28÷174) ×123 = 20. Therefore, 20 were selected in Izzi, 24 in Abakaliki, 21 in Ikwo, 17 in Ezza north, 22 in Afikpo north, and 19 in Afikpo south, giving a total of 123 PHC facilities that were selected. In each of the selected PHC facilities, four participants were selected by balloting from the list of eligible PHC workers. In any facility where the eligible participants were four or less, all were selected.

## Data collection and management

Data was collected via structured self-administered questionnaire survey. The questionnaire design was informed by data published by other studies [24, 31]. The questionnaire was written in English and had 48 items and five sections. The sections were on: Background characteristics of respondents; knowledge of malaria diagnosis and MRDT; opinion about MRDT; availability and use of MRDT; and anti-malarial drug prescription. Expert assessment of the questionnaire validity was done by the research team via a systematic and iterative process of review-discussion-correction-and-redrafting before the final version was pre-tested and minor adjustments made thereafter.

The principal investigator and research assistants distributed questionnaires to participants and, in most cases, waited for the questionnaires to be completed and collected same. Completed questionnaires were crosschecked for consistency and completeness. Questionnaires with inconsistencies and or missing data were returned to the respective respondents for correction. The completed questionnaires were serially and uniquely numbered and data was

entered using Microsoft Excel 2007 (Microsoft Inc., Redmond, Washington) and was verified using Stata (SE) version 15.1 (Stata Corp, College Station, TX, USA).

The outcome measures were the proportion of PHC workers who: (1) usually (routinely) used MRDT for malaria diagnosis (2) used MRDT for malaria diagnosis for all or most (as against some, few, or none) of their patients suspected of having malaria in the preceding month (3) had good anti-malarial drug prescription practice (4) responded appropriately to negative MRDT results by never or rarely (as against sometimes, often, or always) prescribing anti-malarial drugs for patients with negative MRDT results. The PHC workers' anti-malarial drug prescription practice was assessed by scoring their responses to eight questionnaire items. For seven items, each right answer was scored "one" and each wrong answer was scored "zero". For one item, the responses were scored "three" "one" and "zero" as appropriate, giving a maximum score of 10. The total practice score for every respondent was computed and then graded on a two-level scale such that scores of >75% of the maximum attainable score of 10 was taken to be good practice and ≤ 75% as fair or poor practice [31].

The independent factors were age, gender, level of education, occupation or cadre, duration of practice, type of health facility (USAID PMI-supported or not), level of knowledge about malaria diagnosis and MRDT, and level of opinion about the use of MRDT for malaria diagnosis.

## Statistical analysis

The data was analysed using Stata (SE) version 15.1 (Stata Corp, College Station, TX, USA). Frequencies with percentages and means with standard deviations were used to describe the data as appropriate. The associated factors and predictors of the outcome measures were assessed using population-averaged models to account for clustering at the LGA level. For each dichotomous independent factor, prevalence difference in the outcomes with 95% CIs and p-values were computed using binomial identity generalized estimating equations (GEE) with an exchangeable correlation matrix and robust standard errors. For each continuous independent factor, coefficients in the outcomes with 95% CIs and p-values were computed using the binomial identity GEE models. Both bivariate and multivariate GEE models were fitted. Whenever convergence was not achieved by any binomial identity GEE model, a gaussian identity GEE model was used instead [38]. Because of the small number of clusters, the nominal type 1 error rate from the GEE could be inflated due to finite-sample bias. To increase the performance, an adjustment to the GEE model with Wald t-test and Kauermann–Carroll correction via a programme written in Stata was used for finite-sample corrections [39].

## Ethics statement

Ethical approval was obtained from the Research and Ethics Committee of the Alex Ekwueme Federal University Teaching Hospital Abakaliki (Number 10/06/2019–11/06/2019). Written informed consent was obtained from the study participants. The purpose the study, nature of participation, likely duration of participation, voluntary nature of participation, absence of potential harm, potential benefit, and confidential nature of the study were duly communicated to participants.

## Results

### Sociodemographic and background characteristics

As presented in Table 1, the mean age of the 490 study participants was 36.0 years (±8.9). Majority were females (83.9%), were married (76.5%), had tertiary education (70.6%), were

**Table 1. Sociodemographic and background characteristics of the 490 study participants.**

| | n (%) |
|---|---|
| Gender | |
| Male | 79 (16.1) |
| Female | 411 (83.9) |
| Age, years, Mean (±SD) | 36.0 (±8.9) |
| Marital status | |
| Married | 375 (76.5) |
| Divorced or Separated | 3 (0.6) |
| Widowed | 15 (3.1) |
| Single | 97 (19.8) |
| Educational level | |
| Primary | 7 (1.4) |
| Secondary | 137 (28.0) |
| Tertiary | 346 (70.6) |
| Occupation (Cadre) | |
| Health attendant | 147 (30.0) |
| CHEW or CHO | 300 (61.2) |
| Nurse or Midwife | 32 (6.5) |
| Others^ | 11 (2.2) |
| Duration of practice (in years) | |
| $\leq 5$ | 140 (28.6) |
| 6–10 | 126 (25.7) |
| 11–15 | 69 (14.1) |
| 16–20 | 97 (19.8) |
| $\geq 21$ | 58 (11.8) |
| Type of health facility | |
| USAID PMI-Supported | 356 (72.7) |
| Not USAID PMI-supported | 134 (27.3) |
| Strata (senatorial zones) | |
| Ebonyi north | 168 (34.3) |
| Ebonyi central | 158 (32.2) |
| Ebonyi south | 164 (33.5) |

SD = Standard deviation. CHEW = Community Health Extension Workers. CHO = Community Health Officers.
USAID PMI = United States Agency for International Development President's Malaria Initiative.
^Include Environmental Health Officers, Environmental Health Technicians, Medical Lab Technicians.

community health extension workers (CHEWs) or community health officers (CHOs) (61.2%), and were working at health facilities supported by the United States Agency for International Development President's Malaria Initiative (USAID PMI-supported health facilities) (72.7%). A higher proportion of 28.6% had practiced for up to five years or less, 34.3% were in Ebonyi north, 33.5% in Ebonyi south, and 32.2% in Ebonyi central.

## Use of malaria rapid diagnostic test (MRDT) for malaria diagnosis

The results for the 490 study participants are presented in Table 2. Of the 490, 93.1% had ever used MRDT to diagnose malaria, 81.4% usually (routinely) used MRDT for malaria diagnosis and 18.6% usually (routinely) used only clinical symptoms. In the preceding month, 44.5% of

**Table 2. Use of malaria rapid diagnostic test for malaria diagnosis and anti-malarial drug prescription practices by the 490 study participants.**

|  | n (%) |
|---|---|
| **Use of malaria rapid diagnostic test for malaria diagnosis** |  |
| Availability of MRDT kits in health facility of participant |  |
| Have ever had MRDT kits | 455 (92.9) |
| Currently (at the moment) have MRDT kits | 392 (80.0) |
| Use of MRDT for malaria diagnosis by participants |  |
| Have ever used MRDT to diagnose malaria | 456 (93.1) |
| Usually (routinely) use MRDT for malaria diagnosis | 399 (81.4) |
| Usually (routinely) use only clinical symptoms for malaria diagnosis | 91 (18.6) |
| Among their patients suspected of having malaria in the preceding month, used MRDT for malaria diagnosis for: |  |
| ALL the patients | 218 (44.5) |
| MOST of the patients | 164 (33.5) |
| SOME of the patients | 23 (4.7) |
| FEW of the patients | 23 (4.7) |
| NONE of the patients | 62 (12.6) |
| **Anti-malarial drug prescription practices** |  |
| Gave anti-malarial drugs to patients mostly based on: |  |
| Malaria RDT (MRDT) | 399 (81.4) |
| Clinical diagnoses or symptoms | 91 (18.6) |
| Anti-malarial drug mostly used to treat malaria patients: |  |
| ACT | 480 (98.0) |
| Artesunate | 4 (0.8) |
| SP/Fansidar/Maloxine/Amalar | 4 (0.8) |
| Quinine | 2 (0.4) |
| The ACT most frequently used to treat malaria patients: |  |
| Artemeter/Lumefantrine | 468 (95.5) |
| Artesunate/Amodiaquine | 19 (3.9) |
| Artesunate/SP | 3 (0.6) |
| Gave the correct dosage of Artemeter/Lumefantrine | 485 (99.0) |
| Usually gave the following drugs (apart from PCM/ibuprofen) to patients with positive MRDT results: |  |
| Only anti-malarial | 324 (66.1) |
| Anti-malarial and antibiotics | 166 (33.9) |
| Usually gave the following drugs (apart from PCM/ibuprofen) to patients with negative MRDT results: |  |
| Only anti-malarial | 2 (0.4) |
| Anti-malarial and antibiotics | 55 (11.2) |
| Only antibiotics | 239 (48.8) |
| Vitamin C | 1 (0.2) |

*(Continued)*

**Table 2.** (Continued)

| | n (%) |
|---|---|
| No drugs | 193 (39.4) |
| Gave anti-malarial drugs to patients with negative MRDT results: | |
| Never | 239 (48.8) |
| Rarely | 94 (19.2) |
| Sometimes | 125 (25.5) |
| Often | 31 (6.3) |
| Always | 1 (0.2) |

ACT = Artemisinin-based combination therapy. PCM = Paracetamol. SP = Sulphadoxine-pyremethamine

the participants used MRDT for malaria diagnosis for all their patients suspected of having malaria while 33.5% used MRDT for most, 4.7% used MRDT for some, 4.7% used MRDT for few and 12.6% used MRDT for none.

## Anti-malarial drug prescription practices

As presented in Table 2 for the 490 study participants, 98.0% mostly used ACT to treat malaria and 95.5% mostly used Artemeter-Lumefantrine. Apart from paracetamol or ibuprofen: 66.1% usually gave only anti-malarial drugs for positive MRDT results while 33.9% usually gave anti-malarial drugs plus antibiotics; and 48.8% usually gave only antibiotics for negative MRDT results while 11.2% usually gave anti-malarial drugs plus antibiotics. Following negative MRDT results, 48.8% of participants never gave anti-malarial drugs, 19.2% rarely gave, 25.5% sometimes gave, 6.3% often gave, and 0.2% always gave.

## Predictors of the use of malaria rapid diagnostic test for malaria diagnosis

The associated factors and predictors of the use of MRDT for malaria diagnosis are presented in Tables 3 and 4. The adjusted results show that working in a USAID PMI-supported health facility was a predictor of the usual (routine) use of MRDT for malaria diagnosis (adjusted prevalence difference (aPD) 64.0%, 95% CI 28.8–99.3, p<0.001) (Table 3) and of the use of MRDT for malaria diagnosis for all or most of the patients suspected of having malaria in the preceding month (aPD 63.8%, 31.4–96.2, p<0.001) (Table 4).

## Predictors of anti-malarial drug prescription practices

The associated factors/predictors of the level of anti-malarial drug prescription practice and appropriate response to negative MRDT results are respectively presented in Tables 5 and 6.

As shown by the adjusted results in Table 5, the predictors of good anti-malarial drug prescription practice were: having good opinion about the use of MRDT for malaria diagnosis (aPD 29.7%, 13.6–45.9, p<0.001); having good knowledge of malaria diagnosis and MRDT (aPD 10.1%, 3.0–17.1, p = 0.005); being a health attendant (aPD 15.5%, 2.1–28.8, p = 0.023); working in a USAID PMI-supported health facility (aPD 14.5%, 1.4–27.6, p = 0.030); and age as one year increase in age increased the probability of having good anti-malarial drug prescription practice by 0.9% (adjusted coefficient (aCoef) 0.9%, 95% CI 0.2–1.5, p = 0.011). The adjusted results in Table 6 show that having good opinion about the use of MRDT for malaria

**Table 3. Predictors of the usual (routine) use of malaria rapid diagnostic test for malaria diagnosis among the 490 study participants.**

| | Usually (routinely) used MRDT for malaria diagnosis | | Crude results* | | Adjusted results** | |
|---|---|---|---|---|---|---|
| | Yes n (%) 399 (81.4) | No^ n (%) 91 (18.6) | cPD (95% CI) or cCoef (95% CI) | p value | aPD (95% CI) or aCoef (95% CI) | p value |
| Gender | | | | | | |
| Female | 330 (80.3) | 81 (19.7) | 0 | | 0 | |
| Male | 69 (87.3) | 10 (12.7) | 8.4% (-10.0–26.7) | 0.189 | 2.5% (-5.1–10.1) | 0.516 |
| Age, years (coefficient) | – | – | 0.3% (-0.7–1.3) | 0.324 | 0.04% (-0.5–0.6) | 0.873 |
| Educational level | | | | | | |
| Primary or Secondary | 113 (78.5) | 31 (21.5) | 0 | | 0 | |
| Tertiary | 286 (82.7) | 60 (17.3) | -0.6% (-15.8–14.5) | 0.872 | 2.5% (-1.9–6.9) | 0.266 |
| Occupation (Cadre) | | | | | | |
| Health attendant | 116 (78.9) | 31 (21.1) | 0 | | 0 | |
| Others^^ | 283 (82.5) | 60 (17.5) | -1.1% (-16.4–14.3) | 0.794 | -9.1% (-19.1–0.9) | 0.073 |
| Duration of practice, years | | | | | | |
| < = 10 | 217 (81.6) | 49 (18.4) | 0 | | 0 | |
| > = 11 | 182 (81.3) | 42 (18.7) | 3.4% (-5.0–11.7) | 0.224 | 3.9% (-2.2–9.9) | 0.208 |
| Type of health facility | | | | | | |
| Not USAID PMI-Supported | 51 (38.1) | 83 (61.9) | 0 | | 0 | |
| USAID PMI-Supported | 348 (97.8) | 8 (2.2) | 64.3% (-14.7–143) | 0.073 | 64.0% (28.8–99.3) | **< 0.001** |
| Knowledge of malaria diagnosis and MRDT$ | | | | | | |
| Fair or Poor | 162 (75.0) | 54 (25.0) | 0 | | 0 | |
| Good | 237 (86.5) | 37 (13.5) | 7.3% (-14.2–28.9) | 0.280 | 5.3% (-0.7–11.4) | 0.084 |
| Opinion about the use of MRDT for malaria diagnosis$$ | | | | | | |
| Fair or Poor | 102 (75.6) | 33 (24.4) | 0 | | 0 | |
| Good | 297 (83.7) | 58 (16.3) | 4.6% (-16.4–25.7) | 0.443 | 3.0% (-3.6–9.6) | 0.370 |

cPD = Crude prevalence difference. aPD = Adjusted prevalence difference. cCoef = Crude coefficient. aCoef = Adjusted coefficient.

*Adjusted for clustering and stratification.

**Adjusted for clustering, stratification and other independent factors or potential covariates (gender, age, educational level, occupation or cadre, duration of practice, type of health facility, knowledge of malaria diagnosis and MRDT, opinion about the use of MRDT for malaria diagnosis and strata).

CHEW = Community Health Extension Workers. CHO = Community Health Officers. USAID PMI = United States Agency for International Development Presidential Malaria Initiative.

^Usually (routinely) use only clinical symptoms for malaria diagnosis.

^^Include Community Health Extension Workers and Community Health Officers, Nurses and Midwives, Environmental Health Officers, Environmental Health Technicians, Medical Lab Technicians

$Knowledge scores of >75% of maximum of 22 was good knowledge, < = 75% was fair or poor knowledge.

$$Opinion scores of >75% of the maximum of 6 was good opinion, < = 75% was fair or poor opinion.

diagnosis was a predictor of appropriate response to negative MRDT results (never or rarely prescribing anti-malarial drugs) (aPD 44.4%, 28.1–60.8, p<0.001).

## Discussion

This study assessed the use of MRDT for malaria diagnosis and anti-malarial drug prescription practices and the predictors. The descriptive findings show that 93.1% of the PHC workers in Ebonyi state, Nigeria, had used MRDT to diagnose malaria, 81.4% usually (routinely) used MRDT for malaria diagnosis (as against 18.6% that usually used only clinical symptoms), and

**Table 4. Predictors of the use of malaria rapid diagnostic test for malaria diagnosis in the preceding month among the 490 study participants.**

| | Among the patients suspected of having malaria in the preceding month, used MRDT for: | | Crude results* | | Adjusted results** | |
| --- | --- | --- | --- | --- | --- | --- |
| | All or Most n (%) 382 (78.0) | None, Few or Some n (%) 108 (22.0) | cPD (95% CI) or cCoef (95% CI) | p value | aPD (95% CI) or aCoef (95% CI) | p value |
| Gender | | | | | | |
| Female | 313 (76.2) | 98 (23.8) | 0 | | 0 | |
| Male | 69 (87.3) | 10 (12.7) | 8.4% (-12.7–29.5) | 0.227 | 6.5% (-1.7–14.7) | 0.119 |
| Age, years (coefficient) | – | – | 0.2% (-0.6–1.1) | 0.344 | 0.1% (-0.5–0.7) | 0.777 |
| Educational level | | | | | | |
| Primary or Secondary | 108 (75.0) | 36 (25.0) | 0 | | 0 | |
| Tertiary | 274 (79.2) | 72 (20.8) | 0.4% (-16.6–17.4) | 0.932 | 2.6% (-5.9–11.0) | 0.555 |
| Occupation (Cadre) | | | | | | |
| Health attendant | 111 (75.5) | 36 (24.5) | 0 | | 0 | |
| Others^ | 271 (79.0) | 72 (21.0) | -0.2% (-18.5–18.2) | 0.972 | -8.3% (-21.7–5.1) | 0.223 |
| Duration of practice, years | | | | | | |
| < = 10 | 208 (78.2) | 58 (21.8) | 0 | | 0 | |
| > = 11 | 174 (77.7) | 50 (22.3) | 2.8% (-2.8–8.5) | 0.165 | 2.7% (-4.9–10.3) | 0.486 |
| Type of health facility | | | | | | |
| Not USAID PMI-Supported | 47 (35.1) | 87 (64.9) | 0 | | 0 | |
| USAID PMI-Supported | 335 (94.1) | 21 (5.9) | 64.4% (-8.9–138) | 0.063 | 63.8% (31.4–96.2) | **< 0.001** |
| Knowledge of malaria diagnosis and MRDT$ | | | | | | |
| Fair or Poor | 154 (71.3) | 62 (28.7) | 0 | | 0 | |
| Good | 228 (83.2) | 46 (16.8) | 9.3% (-13.0–31.6) | 0.215 | 6.2% (-0.2–12.5) | 0.056 |
| Opinion about the use of MRDT for malaria diagnosis$$ | | | | | | |
| Fair or Poor | 98 (72.6) | 37 (27.4) | 0 | | 0 | |
| Good | 284 (80.0) | 71 (20.0) | 5.2% (-18.5–28.9) | 0.446 | 2.2% (-6.6–11.0) | 0.627 |

cPD = Crude prevalence difference. aPD = Adjusted prevalence difference. cCoef = Crude coefficient. aCoef = Adjusted coefficient.

*Adjusted for clustering and stratification.

**Adjusted for clustering, stratification and other independent factors or potential covariates (gender, age, educational level, occupation or cadre, duration of practice, type of health facility, knowledge of malaria diagnosis and MRDT, opinion about the use of MRDT for malaria diagnosis and strata).

CHEW = Community Health Extension Workers. CHO = Community Health Officers. USAID PMI = United States Agency for International Development Presidential Malaria Initiative.

^Include Community Health Extension Workers and Community Health Officers, Nurses and Midwives, Environmental Health Officers, Environmental Health Technicians, Medical Lab Technicians.

$Knowledge scores of >75% of maximum of 22 was good knowledge, < = 75% was fair or poor knowledge.

$$Opinion scores of >75% of the maximum of 6 was good opinion, < = 75% was fair or poor opinion.

78.0% used MRDT for malaria diagnosis for all or most of their patients suspected of having malaria in the preceding month. Similar findings were reported by other studies in Zamfara state where 82.7% of the health workers routinely used MRDT to diagnose malaria [31], in Ogun state where 85.2% of the public healthcare workers used MRDT for malaria diagnosis [23] and in Ghana where 84% of the health workers routinely used malaria tests for malaria diagnosis [40].

**Table 5. Predictors of the level of anti-malarial drug prescription practice among the 490 study participants.**

| | Practice level^ | | Crude results* | | Adjusted results** | |
|---|---|---|---|---|---|---|
| | Good n (%) 367 (74.9) | Fair or Poor n (%) 123 (25.1) | cPD (95% CI) or cCoef (95% CI) | p value | aPD (95% CI) or aCoef (95% CI) | p value |
| Gender | | | | | | |
| Female | 304 (74.0) | 107 (26.0) | 0 | | 0 | |
| Male | 63 (79.6) | 16 (20.2) | 4.3% (-11.1–19.7) | 0.353 | -3.5% (-12.7–5.7) | 0.454 |
| Age, years (coefficient) | – | – | 0.7% (0.3–1.1) | **0.020** | 0.9% (0.2–1.5) | **0.011** |
| Educational level | | | | | | |
| Primary or Secondary | 111 (77.1) | 33 (22.9) | 0 | | 0 | |
| Tertiary | 256 (74.0) | 90 (26.0) | 0.6% (-8.3–9.4) | 0.810 | 6.7% (-11.0–24.3) | 0.459 |
| Occupation (Cadre) | | | | | | |
| Health attendant | 114 (77.6) | 33 (22.4) | 0.1% (-9.5–9.7) | 0.976 | 15.5% (2.1–28.8) | **0.023** |
| Others^^ | 253 (73.8) | 90 (26.2) | 0 | | 0 | |
| Duration of practice, years | | | | | | |
| < = 10 | 191 (71.8) | 75 (28.2) | 0 | | 0 | |
| > = 11 | 176 (78.6) | 48 (21.4) | 3.3% (-3.6–10.3) | 0.177 | -4.2% (-12.6–4.1) | 0.318 |
| Type of health facility | | | | | | |
| Not USAID PMI-Supported | 85 (63.4) | 49 (36.6) | 0 | | 0 | |
| USAID PMI-Supported | 282 (79.2) | 74 (20.8) | 19.1% (-4.2–42.4) | 0.072 | 14.5% (1.4–27.6) | **0.030** |
| Knowledge of malaria diagnosis and MRDT$ | | | | | | |
| Fair or Poor | 141 (65.3) | 75 (34.7) | 0 | | 0 | |
| Good | 226 (82.5) | 48 (17.5) | 17.1% (-8.3–42.5) | 0.102 | 10.1% (3.0–17.1) | **0.005** |
| Opinion about the use of MRDT for malaria diagnosis$$ | | | | | | |
| Fair or Poor | 73 (54.1) | 62 (45.9) | 0 | | 0 | |
| Good | 294 (82.8) | 61 (17.2) | 30.9% (-8.6–70.4) | 0.078 | 29.7% (13.6–45.9) | **< 0.001** |

cPD = Crude prevalence difference. aPD = Adjusted prevalence difference. cCoef = Crude coefficient. aCoef = Adjusted coefficient.

*Adjusted for clustering and stratification.

**Adjusted for clustering, stratification and other independent factors or potential covariates (gender, age, educational level, occupation or cadre, duration of practice, type of health facility, knowledge of malaria diagnosis and MRDT, opinion about the use of MRDT for malaria diagnosis and strata).

CHEW = Community Health Extension Workers. CHO = Community Health Officers. USAID PMI = United States Agency for International Development Presidential Malaria Initiative.

^Practice scores of >75% of the maximum of 10 was good practice, < = 75% was fair or poor practice.

^^Include Community Health Extension Workers and Community Health Officers, Nurses and Midwives, Environmental Health Officers, Environmental Health Technicians, Medical Lab Technicians.

$Knowledge scores of >75% of maximum of 22 was good knowledge, < = 75% was fair or poor knowledge.

$$Opinion scores of >75% of the maximum of 6 was good opinion, < = 75% was fair or poor opinion.

However, smaller proportions were reported by a study in Rivers state where only 70.4% of PHC workers had used MRDT to diagnose malaria, 39.9% used it almost every day and 30.5% used it for all fever cases [24]. These contrasting lower levels of MRDT use might be due to the fact that the study was conducted some years earlier when the availability and use of MRDT were generally less. The smaller study population in the study (202 PHC workers across only 17 PHC facilities in two LGAs) might also be a contributing factor.

The public health implications of these findings is that although the majority of the PHC workers in Ebonyi state, Nigeria, usually used MRDT for malaria diagnosis, it was still far short of the recommended universal diagnostic testing before treatment [2, 6, 12, 13] as many

**Table 6. Predictors of appropriate response to negative malaria rapid diagnostic test results among the 490 study participants.**

| | Appropriate response^ n (%) 333 (68.0) | Inappropriate response^ n (%) 157 (32.0) | Crude results* | | Adjusted results** | |
|---|---|---|---|---|---|---|
| | | | cPD (95% CI) or cCoef (95% CI) | p value | aPD (95% CI) or aCoef (95% CI) | p value |
| Gender | | | | | | |
| Female | 277 (67.4) | 134 (32.6) | 0 | | 0 | |
| Male | 56 (70.9) | 23 (29.1) | 5.7% (-19.6–31.0) | 0.434 | -4.8% (-20.4–10.7) | 0.542 |
| Age, years (coefficient) | – | – | 0.4% (-0.8–1.6) | 0.268 | 0.5% (-0.1–1.1) | 0.124 |
| Educational level | | | | | | |
| Primary or Secondary | 96 (66.7) | 48 (33.3) | 0 | | 0 | |
| Tertiary | 237 (68.5) | 109 (31.5) | 4.3% (-13.3–21.8) | 0.406 | -18.4% (-59.4–22.7) | 0.380 |
| Occupation (Cadre) | | | | | | |
| Health attendant | 97 (66.0) | 50 (34.0) | 0 | | 0 | |
| Others^^ | 236 (68.8) | 107 (31.2) | 5.5% (-16.7–27.6) | 0.400 | 16.4% (-29.1–61.9) | 0.481 |
| Duration of practice, years | | | | | | |
| < = 10 | 174 (65.4) | 92 (34.6) | 0 | | 0 | |
| > = 11 | 159 (71.0) | 65 (29.0) | 2.5% (-20.4–25.5) | 0.681 | 0.1% (-9.5–9.8) | 0.976 |
| Type of health facility | | | | | | |
| Not USAID PMI-Supported | 83 (61.9) | 51 (38.1) | 0 | | 0 | |
| USAID PMI-Supported | 250 (70.2) | 106 (29.8) | 8.1% (-20.9–37.1) | 0.351 | 5.2% (-0.6–10.9) | 0.078 |
| Knowledge of malaria diagnosis and MRDT$ | | | | | | |
| Fair or Poor | 125 (57.9) | 91 (42.1) | 0 | | 0 | |
| Good | 208 (75.9) | 66 (24.1) | 17.9% (-10.4–46.1) | 0.113 | 8.2% (-1.6–17.9) | 0.100 |
| Opinion about the use of MRDT for malaria diagnosis$$ | | | | | | |
| Fair or Poor | 48 (35.6) | 87 (64.4) | 0 | | 0 | |
| Good | 285 (80.3) | 70 (19.7) | 45.5% (5.6–85.4) | **0.039** | 44.4% (28.1–60.8) | **< 0.001** |

cPD = Crude prevalence difference. aPD = Adjusted prevalence difference. cCoef = Crude coefficient. aCoef = Adjusted coefficient.

*Adjusted for clustering and stratification.

**Adjusted for clustering, stratification and other independent factors or potential covariates (gender, age, educational level, occupation or cadre, duration of practice, type of health facility, knowledge of malaria diagnosis and MRDT, opinion about the use of MRDT for malaria diagnosis and strata).

CHEW = Community Health Extension Workers. CHO = Community Health Officers. USAID PMI = United States Agency for International Development Presidential Malaria Initiative.

^Never or rarely prescribing anti-malarial drugs was appropriate response while sometimes, often or always prescribing anti-malarial drugs was inappropriate response.

^^Include Community Health Extension Workers and Community Health Officers, Nurses and Midwives, Environmental Health Officers, Environmental Health Technicians, Medical Lab Technicians.

$Knowledge scores of >75% of maximum of 22 was good knowledge, < = 75% was fair or poor knowledge.

$$Opinion scores of >75% of the maximum of 6 was good opinion, < = 75% was fair or poor opinion.

of them still usually used only clinical symptoms to diagnose malaria. This indicates over-diagnosis and over-treatment of malaria with heightened risk of selection pressure and drug resistance. To significantly reduce over-diagnosis and overtreatment of malaria, all patients suspected of having malaria should receive diagnostic testing with MRDT (or microscopy) before treatment with anti-malarial drugs. Although the availability and use of MRDT has increased over the past decade due to foreign partners collaboration with the Nigerian and Ebonyi state governments, this study findings underscore the need for more efforts and interventions in the drive to achieve universal diagnostic testing before malaria treatment.

Findings of this study also show that 74.9% of the PHC workers had good anti-malarial drug prescription practice. While 98.0% of the PHC workers mostly used ACT to treat malaria patients, 95.5% of them mostly used artemether-lumefantrine. Similarly, in a study in Sudan ACT was the most frequently prescribed anti-malarial drug among public PHC workers, however, it was among a slightly lower proportion of 88.4% of the health workers and artesunate plus sulphadoxine-pyrimethamine was the mostly used ACT [41]. Also, another study in India reported that majority (63.6%) of public practitioners always prescribed chloroquine while only 14.1% always prescribed artesunate plus sulphadoxine-pyrimethamine (an ACT) [42]. The study identified inadequate availability of the recommended ACT, inertia of practitioners in accepting new treatments, lack of awareness about the new drug policy, and fear of stock outs as reasons for this pattern of prescription [42]. This explains the difference in the prescription practice compared to our study where many PHC facilities were being supported by USAID PMI to provide malaria treatment with ACT, particularly artemether-lumefantrine, according to policy recommendations.

In this study, 66.1% of the PHC workers usually prescribed only anti-malarial drugs (in addition to paracetamol/ibuprofen) for patients with positive MRDT results as against 33.9% that usually prescribed anti-malarial drugs and antibiotics. A lower value was reported by a study in Sudan where 24.2% of prescriptions for malaria positive patients contained unexplained antibiotic (without a clear indication for it) [41]. From public health perspective, even though majority of the PHC workers in Ebonyi state had good anti-malarial drug prescription practice, the unnecessary co-prescription of anti-malarial drugs and antibiotics for patients with positive MRDT results was relatively high (was usually practiced by about a third of them). This high rate of unnecessary co-prescription of antibiotics would lead to wastage, increased cost of treatment, more side effects, and increased risk of the development of drug (antibiotics) resistance. This calls for more interventions to improve the anti-malarial drug prescription practices of the PHC workers in Ebonyi state. Qualitative studies are also needed to provide more insights on this high rate of unnecessary co-prescription of antibiotics.

Regarding appropriate response to negative MRDT results in this study, 68.0% of the PHC workers reported that they never or rarely prescribed anti-malarial drugs for patients with negative MRDT results (appropriate response) while 32.0% sometimes or often or always prescribed anti-malarial drugs (inappropriate response). Similarly, in a study across six states (in five geopolitical zones) in Nigeria 32% of the health workers reported they would prescribe ACT for patients with negative MRDT results [27]. A slightly lower value was reported by another study in Oyo state where 26% of the health workers prescribed anti-malarial drugs to patients with negative MRDT results [28].

However, higher proportion of 81% was reported by a study in Sokoto regarding good adherence to MRDT results (prescribing ACT for patients with positive MRDT results and withholding ACT for patients with negative MRDT results) [43]. Also, in a study in Tanzania a higher proportion of 67% of PHC workers reported that they sometimes prescribed anti-malarial drugs to MRDT negative patients [44] but the higher value might be because that study involved only 18 health workers and was conducted some years earlier and just two years after MRDT was introduced in the study area (at a time when the confidence in MRDT result was relatively low). Also, in a study in Ghana only 9% of health workers reported that they always adhered (100%) with malaria test results in managing their patients while 60% reported that they adhered 80–99% [40].

Compared with the 68.0% in this study, a much lower proportion of 27.5% of the medical doctors in Ebonyi state were reported to never or rarely prescribed anti-malarial drugs for patients with negative MRDT results [45]. This much lower value could be due to the fact that the medical doctors (across tertiary and secondary health facilities), in most cases, believed in

using their clinical judgement and discretion with less compliance to the recommendation that anti-malarial drugs should be prescribed for only patients with positive MRDT (or microscopy) results. This was in contrast to the PHC workers who practiced based on standing orders, with less room for the use of discretion. It is worth noting that the recommendations in diagnosis and treatment guidelines are evidence-based. MRDT is not only much more feasible to deploy and use by PHC workers in remote rural settings (compared to microscopy), it is also accurate in diagnosing malaria. The diagnostic accuracy of quality assured MRDT (and the effectiveness in targeting anti-malarial drugs and antibiotics when negative results are adhered to) has been reported to be far more than that of presumptive diagnosis and comparable to good field/routine microscopy [6, 46] and expert microscopy [47]. Moreover, MRDT is more cost-effective compared to microscopy (and presumptive diagnosis) [48] especially when providers and patients adhere to test results [29]. There is thus the need for more sensitization and re-orientation of all categories of relevant healthcare professionals in the foregoing regard.

The above study findings have public health significance. Even though majority of the PHC workers had good anti-malarial drug prescription practice, many of them (about a third) still prescribed anti-malarial drugs for patients with negative MRDT results more frequently than necessary. The practice of prescribing anti-malarial drugs for patients with negative MRDT results further add to the problem of over-diagnosis and over-treatment that results from presumptive diagnosis (non-use of MRDT or microscopy) as discussed earlier. The implication of this is that even if a very high rate of diagnostic testing with MRDT is achieved, the problem of over-treatment of malaria and risk of drug resistance will still remain if many health workers continue to prescribe anti-malarial drugs (including ACT) for patients with negative MRDT results. There is therefore the need for more public health interventions in this regard.

This study identified some predictors of the use of MRDT and anti-malarial drug prescription practices among the PHC workers. Working at USAID PMI-supported health facilities was a strong predictor of the use of MRDT for malaria diagnosis. This was perhaps because USAID PMI supported/supplied these facilities regularly with malaria commodities including MRDT kits which were thus always available and also free of charge to patients. This enhanced the use of MRDT for malaria diagnosis. This contrasted with those facilities not supported by USAID PMI which had to buy MRDT kits from the market with implications on availability of kits and affordability by patients. Patients not having to pay for MRDT was reported as a predictor of utilization of MRDT by health workers in Zamfara state [31]. These findings emphasize the need for the Ebonyi state government with her foreign partners to extend the regular supply of MRDT kits to the PHC facilities not receiving USAID PMI support in order to facilitate the achievement of universal diagnostic testing before malaria treatment.

Although PHC workers with good knowledge of malaria diagnosis and MRDT and those with good opinion about the use of MRDT for malaria diagnosis were more likely to use MRDT, these factors were not predictors. However, a study in Zamfara state reported that having good knowledge of MRDT and trust in MRDT results were predictors of MRDT utilization by health workers [31]. This contrasting finding could perhaps be explained by the different analytic approaches as our study, unlike the previous study, used recommended analytic techniques that minimized the nominal type 1 error rate from clustering and finite-sample bias.

The predictors of good anti-malarial drug prescription practice identified by our study included, among others, having good opinion about the use of MRDT for malaria diagnosis, having good knowledge of malaria diagnosis and MRDT and working in USAID PMI-supported health facilities. We did not identify any comparable study that assessed the level of anti-malarial drug prescription practice and the predictors. However, for the same reason as explained above, working at USAID PMI-supported health facilities (compared to the non-supported facilities) was a predictor of good prescription practice because the recommended

anti-malarial drug, ACT and particularly artemether-lumefantrine, were always available and free for malaria patients. This perhaps partly enhanced good prescription practice. Another reason might be due to more regular USAID-PMI and SMEP led trainings and supportive supervision for the supported facilities regarding the recommendation of universal diagnostic testing before treatment. Since effective management and control of malaria requires not only use of diagnostic testing (such as MRDT) by PHC workers but also their level of anti-malarial drug prescription practice, the above predictors provide important guidance for malaria management and control interventions in Ebonyi state and similar settings.

Having good opinion about the use of MRDT for malaria diagnosis was a strong predictor of appropriate response (never or rarely prescribing anti-malarial drugs) by PHC workers following negative MRDT results. This might be due to their confidence or trust in MRDT results and their beliefs in the accuracy and reliability of MRDT. Similarly, poor perception of MRDT was a predictor of ACT prescription by health workers for negative MRDT in the preceding six months in another study in Ebonyi state [49]. Having good knowledge of malaria diagnosis and MRDT was not a predictor in this study as it insignificantly increased the rate of appropriate response following negative MRDT results. However, a study in Oyo state reported health workers' poor knowledge of the causes of fever as a predictor of the prescription of anti-malarial drugs to patients with negative MRDT results [28]. But it should be noted, as explained above, that our analytic techniques were quite different.

In this study, working in USAID PMI-supported health facilities insignificantly increased the rate of appropriate response following negative MRDT results and was not identified as a predictor. The fact that we identified working in USAID PMI-supported health as a predictor of use of MRDT and of good anti-malaria drug prescription practice but not of appropriate response following negative MRDT results has important implications. This finding implies that although MRDT and ACT were perhaps always available and affordable and trainings with supportive supervisions more regular at USAID PMI-supported health facilities to enhance the use of MRDT and overall anti-malaria drug prescription practice among the PHC workers, more innovative strategies are needed to enhance their appropriate response following negative MRDT results, including strategies to enhance their opinions about the use of MRDT for malaria diagnosis (the only predictor identified by this study) and further studies are imperative in this regard. Specifically, qualitative studies are also needed to provide more insights about factors that influence PHC workers' response to negative MRDT results in the context of increased availability and use of MRDT in low resource settings.

A limitation in this study was the use of questionnaires to collect self-reported data about practices which relied on the respondents providing the right data. It was thus subject to reporting bias as there was the tendency for respondents to overestimate desirable outcomes and or underestimate undesirable outcomes. However, the bias was prevented or minimized by making the questionnaire anonymous and by assuring respondents of and ensuring high degree of confidentiality.

## Conclusions

This study has shown that majority of the PHC workers in Ebonyi state, Nigeria, used MRDT usually (routinely) for malaria diagnosis and used MRDT to diagnose malaria for all or most of their suspected malaria patients in the preceding month. However, the use of MRDT for malaria diagnosis among the PHC workers still fell far short of the universal diagnostic testing recommended by the WHO and the Nigerian National Guidelines for Diagnosis and Treatment of Malaria as many still usually used presumptive diagnosis. Majority of the PHC workers had good anti-malarial drug prescription practice, however, many of them unnecessarily

usually co-prescribe anti-malarial drugs and antibiotics for patients with positive MRDT results and many still prescribed anti-malarial drugs for patients with negative MRDT results. The predictors identified were: working in USAID PMI-supported health facilities (for the use of MRDT for malaria diagnosis); having good opinion about the use of MRDT for malaria diagnosis, having good knowledge of malaria diagnosis and MRDT, being a health attendant, working in USAID PMI-supported health facilities and increase in age (for good anti-malarial drug prescription practice); and having good opinion about the use of MRDT for malaria diagnosis (for appropriate response to negative MRDT results).

The evidence in this study emphasize the need for, and highlight factors that should be considered by, further policies actions and interventions to improve the use of MRDT towards universal diagnostic testing and anti-malarial drug prescription practices. It particularly calls for more innovative strategies to enhance appropriate response following negative MRDT results among the PHC workers in Ebonyi state, Nigeria, and similar settings.

## Supporting information

**S1 Dataset.**
(XLSX)

## Author Contributions

**Conceptualization:** Ugwu I. Omale.

**Data curation:** Ugwu I. Omale.

**Formal analysis:** Ugwu I. Omale.

**Investigation:** Ugwu I. Omale.

**Methodology:** Ugwu I. Omale, Benedict N. Azuogu, Adaoha P. Agu, Edmund N. Ossai.

**Project administration:** Ugwu I. Omale.

**Resources:** Ugwu I. Omale.

**Validation:** Ugwu I. Omale, Benedict N. Azuogu, Adaoha P. Agu, Edmund N. Ossai.

**Visualization:** Ugwu I. Omale.

**Writing – original draft:** Ugwu I. Omale.

**Writing – review & editing:** Ugwu I. Omale, Benedict N. Azuogu, Adaoha P. Agu, Edmund N. Ossai.

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
