## [Decision Letter · Decision Letter 0]

2 Apr 2024

PONE-D-24-09527Use of malaria rapid diagnostic test and anti-malarial drug prescription practices among primary healthcare workers in Ebonyi state, Nigeria: an analytical cross-sectional studyPLOS ONE

Dear Dr. Omale,

Thank you for submitting your manuscript to PLOS ONE. After careful consideration, we feel that it has merit but does not fully meet PLOS ONE’s publication criteria as it currently stands. Therefore, we invite you to submit a revised version of the manuscript that addresses the points raised during the review process.

We look forward to receiving your revised manuscript.

Kind regards,

Yash Gupta, Ph.D.

Academic Editor

PLOS ONE

Journal Requirements:

a) The name of the colleague or the details of the professional service that edited your manuscript.

b) A copy of your manuscript showing your changes by either highlighting them or using track changes (uploaded as a *supporting information* file).

c) A clean copy of the edited manuscript (uploaded as the new *manuscript* file).

**Additional Editor Comments:**

Authors need to address all comments by the reviewers

Reviewers' comments:

Reviewer's Responses to Questions

**Comments to the Author**

1. Is the manuscript technically sound, and do the data support the conclusions?

Reviewer #1: Yes

Reviewer #2: Yes

2. Has the statistical analysis been performed appropriately and rigorously? 

Reviewer #1: Yes

Reviewer #2: Yes

3. Have the authors made all data underlying the findings in their manuscript fully available?

Reviewer #1: Yes

Reviewer #2: Yes

4. Is the manuscript presented in an intelligible fashion and written in standard English?

Reviewer #1: Yes

Reviewer #2: Yes

5. Review Comments to the Author

Reviewer #1: Title:

Use of malaria rapid diagnostic test and anti-malarial drug prescription practices among primary healthcare workers in Ebonyi state, Nigeria: an analytical cross-sectional study

The article has addresses an insight, in relation to the use of MRDTs to diagnose malaria in Nigeria, there's a gap between policy and practice. Improvements are needed in knowledge, attitudes, and potentially targeted interventions for specific healthcare workers to ensure they strictly follow negative test results and reduce unnecessary prescription of anti-malarial drugs. This will help combat drug resistance and improve overall malaria treatment practices.

Comments:

1. Authors have addressed the importance of Kit based diagnosis and medication based on the results. However, authors should also mention the malarial infection prevalent i.e., the infection due to different species.

2. The methodology section can be written precisely.

sections like Study design, setting and participants: the lines from 130 to 145 can be shortened and many of the lines not relevant to the manuscript may be removed.

Section sample size and sampling techniques calculation part can be avoided and is also not required.

For Statistical analysis: The section can be briefly written as these can be indicated in the table section for better readability.

Authors must also discuss about the efficiency of the MRDT in detection of malaria.

3. All the references are relevant.

4. Authors have stressed on the implementation of better policies to overcome the problems which might arise from overmedication

Reviewer #2: The authors have narrated the gaps and factors that can be improved for the use of MRDT kits and anti-malaria drug prescription. The manuscript is well written and the methods are sufficient to replicate the results or to implement the findings.

6. PLOS authors have the option to publish the peer review history of their article (what does this mean?). If published, this will include your full peer review and any attached files.

Reviewer #1: No

Reviewer #2: **Yes: **Namrata Anand

---

## [Author Response · Author response to Decision Letter 0]

10 May 2024

RESPONSES TO COMMENTS BY REVIEWERS 

Thank you for creating the time to review our manuscript and for your comments which have contributed to the improvement of the manuscript.

Reviewer # 1 

1. Comment: Authors have addressed the importance of Kit based diagnosis and medication based on the results. However, authors should also mention the malarial infection prevalent i.e., the infection due to different species.

Response: The information has been added (Lines 68–71).

2. Comment: The methodology section can be written precisely.

sections like Study design, setting and participants: the lines from 130 to 145 can be shortened and many of the lines not relevant to the manuscript may be removed.

Section sample size and sampling techniques calculation part can be avoided and is also not required.

For Statistical analysis: The section can be briefly written as these can be indicated in the table section for better readability.

Response: 

(i) We understand that the description of the study participants in this section is relevant to the manuscript. However, we have modified some sentences as appropriate.

(ii) The calculation part of the sample size estimation has been removed.

(iii) Please note that the statistical analysis section is just one paragraph, we seem not to understand how to further reduce it.

3. Comment: Authors must also discuss about the efficiency of the MRDT in detection of malaria.

Response: Statements regarding MRDT accuracy/effectiveness and cost-effectiveness has been added to the discussion as appropriate (lines 357–366).

4. Comment: All the references are relevant. 

5. Comment: Authors have stressed on the implementation of better policies to overcome the problems which might arise from overmedication.

Response: We are pleased that the reviewer has good understanding of our study concepts and findings.

Reviewer # 2 

Comment: The authors have narrated the gaps and factors that can be improved for the use of MRDT kits and anti-malaria drug prescription. The manuscript is well written and the methods are sufficient to replicate the results or to implement the findings. 

Response: We are pleased that the reviewer has good comprehension of the concepts and findings of our study.

---

## [Editor Report · Decision Letter 1]

15 May 2024

Use of malaria rapid diagnostic test and anti-malarial drug prescription practices among primary healthcare workers in Ebonyi state, Nigeria: an analytical cross-sectional study

PONE-D-24-09527R1

Dear Dr. Omale,

We’re pleased to inform you that your manuscript has been judged scientifically suitable for publication and will be formally accepted for publication once it meets all outstanding technical requirements.

Kind regards,

Yash Gupta, Ph.D.

Academic Editor

PLOS ONE

Additional Editor Comments (optional):

Author response and the revised manuscript adress all the issues raised by the reviewer. The current revised manuscript is fit for publication in PlosOne
---

## [Editor Report · Acceptance letter]

24 May 2024

PONE-D-24-09527R1 

PLOS ONE

Dear Dr. Omale, 

I'm pleased to inform you that your manuscript has been deemed suitable for publication in PLOS ONE. Congratulations! Your manuscript is now being handed over to our production team.

Kind regards, 

on behalf of

Dr. Yash Gupta 

Academic Editor

PLOS ONE